# Skillful Control of Dispersion and 3D Network Structures: Advances in Functional Organic–Inorganic Nano-Hybrid Materials Prepared Using the Sol-Gel Method

**DOI:** 10.3390/polym14163247

**Published:** 2022-08-09

**Authors:** Hiroki Ikake, Shuta Hara, Shigeru Shimizu

**Affiliations:** 1Department of Materials and Applied Chemistry, College of Science and Technology, Nihon University, 1-8-14 Kandasurugadai, Chiyoda-ku, Tokyo 101-8308, Japan; 2Department of Material and Life Chemistry, Faculty of Engineering, Kanagawa University, 3-6-1 Rokkakubashi, Kanagawa-ku, Yokohama 221-8686, Japan

**Keywords:** sol-gel process, 3D network hybrid materials, nanoparticles, nanodispersity, ionic liquids

## Abstract

Organic–inorganic hybrid materials have become indispensable high-performance and highly functional materials. This is owing to the improved dispersion control in hybrid materials and emergence of functional ionic liquids. Harmonization of both these factors has enabled the utilization of functional 3D network structures and nanodispersions in composite materials. Polymeric materials endow materials with flexibility, toughness, and shape-memory properties, whereas inorganic materials provide materials with unique optical, electrical, and magnetic properties due to their nanosize. Organic–inorganic hybrid materials have evolved into novel materials that go beyond the composite rule. In this review, the historical development of hybrid materials prepared using the sol-gel method and the birth of ionic liquids have been summarized. In addition, the historical results leading to the development of functional 3D network structures and dispersion control have also been presented, as well as a review of the research on functional ionic liquids, which are of current interest. The authors also summarize the results of their research on functional ionic liquids. The design of new organic–inorganic hybrid materials has been discussed and the future prospects of new polymer composite materials provided.

## 1. Birth of Organic–Inorganic Hybrid Materials

Organic polymers are low-weight, flexible, and supple, and have excellent formability and workability. Conversely, inorganic materials have excellent functionality, such as high elasticity and heat resistance, as well as optical, magnetic, and electrical properties. Consequently, organic–inorganic hybrid materials combine the excellent properties of both these materials. By appropriately designing the substances and elements to be combined as well as the way in which they are combined, they can be transformed into unique materials with new and diverse functionalities that go beyond their mere combination [1,2,3,4,5,6,7,8].

Hybrids between organic polymers and inorganic materials that exhibit excellent properties are found in nature. For example, shells, teeth, bones, siliceous algae, and rice are typical organic–inorganic hybrids created in nature via biomineralization, in which the inorganic substances are synthesized in vivo. Plants absorb silica from the soil through their roots, which is then accumulated as amorphous silica (opal) in their stems and leaves [9,10]. Strong wind-resistant and rain-resistant stems are supported by silica. In the leaves, the scattering effect of opal promotes plant photosynthesis, and opal precipitation near the leaf pores regulates the water content. Some plants are sensitive to salt damage. In regions with low rainfall, plants protect themselves from salt damage by actively forming hybrids with high silica contents. Animals too use their abundant calcium deposits. For example, *Trochus niloticus* (Figure 1), which inhabits the seas of Okinawa and other coral reefs around the world, has a densely stacked structure of polygonal aragonite (CaCO_3_) (width ~8 mm; thickness ~0.9 mm; stacking interval ~40 nm) [11].

The structure consists of a planar arrangement of ca. 95 wt.% aragonite tablets and ca. 5 wt.% biopolymers connecting the tablets. This regular arrangement allows one to see the beautiful structural color of the pearls, which is their characteristic luster. The high homogeneity and orderliness of the nacreous layer allows it to develop excellent mechanical properties. In other words, *Trochus niloticus* is protected by an excellent high-performance hybrid material.

The wings of the morpho butterfly emit a brilliant iridescent blue (Figure 2). Nanophotonic structures are built on wing scales in a structure so dense that it is almost mystical. Although the physics of structural color is well-established, replicating wing structures by conventional top-down lithography remains challenging.

Natural organic–inorganic hybrid materials are excellent composites with a dense structure consisting of an organic and an inorganic phase. Elaborate structures, such as homogeneous dispersions, phase separation, and dense structures, endow hybrid materials with functionalities beyond those predicted (composite law). Recently, considerable research attention has been directed toward the development of biomimetic hybrid materials that mimic excellent natural hybrid materials, which are rich in functionality [12,13,14]. In particular, organic–inorganic hybrid materials exhibiting ultrastructures with nanoscale dimensions in the dispersed phase have attracted considerable research interest [15,16,17]. The ultrafine structure of the dispersed phase gives rise to quantum effects that are not observed in the bulk material. The excellent properties of these materials, such as those observed for the aforementioned natural hybrid materials, have led to their widespread application in artificially prepared hybrid materials. In the field of optics, these materials satisfy the transparency, orientation, and wide viewing angles required for flat-screen displays. 

Inorganic materials also exhibit properties that are different from their bulk counterparts, which can be hybridized with polymers to provide various properties, such as gas barrier [18], light modulation [19], and electrical conductivity [7,20]. These properties cannot be obtained from the bulk polymers and are new properties that can only be achieved via the formation of an ultrafine structure with inorganic materials and hybridization on a molecular level. However, the nanometer-sized dispersion of the inorganic materials in the polymer matrix generates extremely high interfacial free energy on the surface of the nanoparticles. This causes the nanoparticles to aggregate in the matrix, which not only causes the nanoparticles to lose their unique properties, but also reduces their suppleness, toughness, and formability. Therefore, the development of methods to produce highly dispersed inorganic nanoparticles in a polymer matrix is important for the future use of nanoparticles in devices.

## 2. Preparation of Hybrid Materials Using the Sol-Gel Method

The sol-gel method can be used to hybridize organic polymers and inorganic substances on a molecular level. The sol-gel method prepares these materials via the gelation of the precursor solution upon the hydrolysis and polycondensation reaction of an alkoxysilane. Because the starting materials are liquids, it can be applied to many organic polymers and various inorganic materials by adjusting the miscibility of the two components. The resulting hybrids show a variety of forms, from networked structures formed between the organic and inorganic phases to simply dispersed microparticles in a polymer matrix [21], which have a significant bearing on the overall functionality of the material.

The sol-gel method has a long history. Metal alkoxides were developed as raw materials to prepare hybrid materials by Ebelmen [22] in 1846. Research using the sol-gel method, which had stagnated for a long time, was triggered by the preparation of sintered polycrystals using metal alkoxides by Mazdiyasni et al. [23] in 1969 and Dislich [24] in 1971. Dislich developed the low-temperature hydrolysis of silicate esters, which is thought to be the beginning of the use of silicate ester hydrolysis as a glass production method. Silica gel was prepared via the hydrolysis of silicate esters. Dislich used the hydrolysis and polycondensation of metal alkoxides to prepare powdered gel particles, which were then hot-pressed to produce transparent PYREX^Ⓡ^ glass. Subsequently, research on the use of various metal alkoxides and the preparation of ceramics began to flourish. The sol-gel method has become a promising technique for hybridization using heat-sensitive organic materials because glasses and ceramics can be synthesized under relatively mild conditions. In particular, the development of hybrid materials comprising organic polymers and inorganic materials, and the development of hard contact lens materials by Philipp and Schmidt [25] in 1984, are regarded as epoch-making events. Mixed solutions containing epoxysilane, methacryloxysilane, and titanium alkoxide were subjected to condensation reactions, to which methacrylate (MMA) monomer was added to obtain an injectable homogeneous reaction solution. The viscous solution was heated and cured at temperatures up to 150 °C to obtain colorless, transparent hybrid materials in which the MMA monomer was polymerized into polymethyl methacrylate (PMMA). The PMMA hybrid has a network structure constructed from siloxane (Si-O-Si) bonds, which is also bonded to titanium (Si-O-Ti) via these bonds. The 3D network structure imparts high tensile strength, flexibility, and toughness to the hybrid. The dimethylsiloxane backbone and presence of alcohol groups in the side chains endow the hybrid material with good oxygen permeability, good wettability to tears, and excellent resistance to protein adhesion.

Organic–inorganic hybrid materials prepared using the sol-gel method have evolved from mere blends of organic polymers and inorganic substances to materials with new functional properties that are neither organic polymers nor inorganic substances due to their combination on a molecular level. Recently, the hybridization of many organic polymers and inorganic materials using the sol-gel method has been extensively carried out and the sol-gel method has attracted considerable interest as a technology used for the development of new materials. As mentioned beforehand, the 3D network structure is very important in regard to the physical properties of hybrid materials, and the silane coupling agents described in the next section significantly contribute to the creation of these 3D structures in a highly controlled manner.

The preparation of hybrid materials comprising organic polymers and metal oxides using the sol-gel method from the perspective of the silane coupling agents used is presented in the next section. The formation of cross-linking between the organic and inorganic phases by the silane coupling agents, the historical development of high-performance organic–inorganic hybrid materials, and the new adverse effects of hybridization are also described.

## 3. Creation of Nano-Hybrids Using Silane Coupling Agents

Recently, the preparation of organic–inorganic hybrid materials using the sol-gel method has attracted interest in the homogeneous dispersion or compositing of inorganic materials in nanometer size in a polymer matrix. However, the nanometer size of these particles results in high interfacial free energy on the surfaces of particles and the particles aggregate to reduce this free energy. In this process, small gaps are created between the agglomerating microparticles. As the polymer chains cannot penetrate these gaps, a difference in the osmotic pressure occurs between the inside and outside of the polymer, resulting in weak agglomeration [26,27] due to the solvent depletion effect [28]. This leads to microscopic phase separation and heterogeneous dispersion in many hybrid materials. The particle surface potential is important for the stable dispersion of the particles. This implies that the dispersion can be controlled using the appropriate design of the surfaces of particles.

The preparation of organic–inorganic hybrid materials using the sol-gel method has been extensively studied by Chujo et al. [29,30,31,32,33]. Using the sol-gel method, organic–inorganic hybrid materials are mainly prepared utilizing the following methods: **(i)** mixing and composite formation of the pre-synthesized organic polymers and inorganic materials, **(ii)** synthesis and composite formation of the organic polymers in the presence of the inorganic materials, and **(iii)** in situ preparation and composite formation of the inorganic nanoparticles in a polymer matrix depending on the type of organic polymer or inorganic material used and the expected material properties. These techniques have been used where appropriate. Recently, method **(iii)** has become the standard because hybridization on a molecular level is often achieved. When preparing silica via the sol-gel method, a silicon alkoxide, such as tetraethoxysilane [Si(OC_2_H_5_)_4_; TEOS] or tetramethoxysilane [Si(OCH_3_)_4_; TMOS], is often used as the precursor. Hydrolysis and polycondensation of TEOS forms silica with a 3D network structure constructed via Si-O-Si bonds, as shown in Figure 3 [34].

Nanometer-sized silica can be dispersed in situ in a polymer matrix when TEOS is mixed with an organic polymer and subjected to the sol-gel reaction. Metal oxide nanoparticles with 3D network structures constructed by M-O-M bonds can be synthesized in situ by changing the starting material from Si to a metal (M) alkoxide [M(OR)_n_; M = Ti, W, Cu, Mg, etc.; R = CH_3_, C_2_H_5_, C_3_H_7_, and C_4_H_9_] [35,36]. According to Sanchez et al. [37], hybrids are classified as Class I or Class II based on the bonding state of the organic and inorganic materials. Class I refers to hybrids in which the inorganic substances are dispersed and combined in the polymer matrix via weak interactions, such as van der Waals forces, hydrogen bonds, and electrostatic forces, whereas Class II refers to hybrids in which the organic polymers and inorganic substances are combined via strong bonds, such as covalent and coordination bonds. Metal–carbon bonds are susceptible to hydrolysis and the bond between the two atoms is weak. Hence, silicon–carbon bonds, which are less susceptible to hydrolysis, are often utilized in Class II hybrids, and organic groups and silane coupling agents incorporating silicon–carbon bonds are often used in these organic–inorganic hybrid materials. In particular, the polymer side chains and termini are often modified, substituted, or copolymerized with silane coupling agents in hybrid materials comprising organic polymers and inorganic materials [38,39,40]. Furthermore, silane coupling agents not only improve the dispersibility of the nanoparticles, but also have excellent features that easily form Si-O-Si or Si-O-M bonds and 3D network structures [41] between the organic and inorganic phases.

The 3D network structure not only gives the hybrid material flexibility, but also contributes to the weather resistance of the material. One of the general-purpose polymers is polyurethane (PU), a generic term for polymers with urethane (-NHCOO-) groups in the main chain structure, obtained by the polyaddition reaction of polyhydric alcohols and isocyanates. Although PU exhibits excellent rubber elasticity due to the pseudo-network structure induced by isocyanates, aromatic isocyanates are vulnerable to light exposure, causing yellowing and oxidative degradation. Ikake et al. [42,43] focused on the network structure of PU and constructed 3D networks with high photostability by using a cross-linking agent instead of aromatic isocyanates. The properties of PU elastomers derive from this network structure, which also offers the advantage that the cross-linking density can be easily adjusted by adding appropriate amounts of metal alkoxides. Both ends of the polyol were modified with silane coupling agents, and titanium tetraisopropoxide was added to generate titania nanoparticles in situ, forming a 3D network within the PU matrix. The silane coupling agent enabled the titania nanoparticles to be highly dispersed in the PU matrix, resulting in hybrids with excellent transparency in the visible light range and UV-protection properties without yellowing. Furthermore, by changing the type of metal alkoxide used, various properties can be imparted to the PU. For example, copper(II) ethoxide (CuOEt) can be used to produce CuO nanoparticles, which do not contribute to the network structure because CuOEt is bifunctional. However, the PU matrix is reinforced by the Si-O-Si bonds originating from the coupling agent and the CuO nanoparticles dispersed in its network mesh (Class I hybrid). This effectively utilizes the band energy of CuO, making the PU hybrid material transparent in the visible light region, but able to block near-infrared (heat) radiation. In the case of tungsten trioxide (WO_3_), which exhibits photochromic properties, WO_3_ forms a tungsten bronze structure [44] upon UV irradiation, giving it a bright bronze-blue color. However, the resulting hybrid properties considerably depend on the method used to form the WO_3_ composite. The choice of the WO_3_ precursor is important because the use of tungsten alkoxide produces Class II hybrids, which do not exhibit photochromic properties. Conversely, Class I hybrids are produced when sodium tungstate(VI) dihydrate is used as the precursor and a WO_3_ solution in which the Na-type is changed to H-type via a cation exchange resin is used. The resulting hybrids show good photochromic properties [45,46]. Leaustic et al. [47] showed that UV irradiation of an aqueous suspension of WO_3_ formed tungsten bronze structures using electron spin resonance spectroscopy. Colton et al. [48] also showed that the d-orbital electrons of tungsten were excited by UV light causing a transition from W(VI) to W(V) with a decrease in energy of 1.2 eV using the X-ray photoelectron spectra of amorphous WO_3_ powder samples. In Class II hybrids, the d-orbital of tungsten contributes to the bonding interactions, which may explain why the transition is less likely to occur and why photochromic properties were not exhibited. This is an example of the effect of the starting material used in the sol-gel method on the materials’ properties.

Furthermore, the sol-gel method is useful for preparing thermosetting resins. The hybridization of polyimide (PI) with silica using the sol-gel method was reported by Morikawa et al. [49] in 1992. Tetraethoxysilane (TEOS) was added to polyamide acid (PAA) to obtain silica particles via in situ thermal cross-linking and dispersed in the PI matrix. The silica content was low because the hybrid is a Class I direct dispersion with a uniform dispersion achieved at < 8 wt.%; the dispersion was non-uniform at higher silica contents. The PI hybrid was completed upon cyclization of the PAA end during the thermal imidization reaction; the effect of modifying both ends of PAA with a silane coupling agent on the thermal imidization reaction was not determined and is unknown. In 1997, Srinivasan et al. [50] synthesized and characterized PAA oligomers bearing trimethoxysilyl groups introduced upon modification by a silane coupling agent. The oligomers were then thermally cured to form a cross-linked network with *T*_g_ > 450 °C and a thermal decomposition temperature of 500 °C. The PAA oligomers show excellent performance as low-viscosity precursors to cross-linked PI networks. Chan et al. [51] reported in 2002 that both ends of PAA can be modified with 3-aminopropyltrimethoxysilane (APrTMOS) and TEOS added to prepare PI-Silica hybrid materials. Silica was homogeneously dispersed by the silane coupling agent and the silica content in the hybrid was increased to 54.9 wt.%. At low silica contents, the hybrids form a 3D network structure constructed by Si-O-Si bonds, whereas, at higher silica contents, more Si-OH residues are formed upon hydrolysis of the alkoxy groups in APrTMOS or TMOS.

In the field of electronic materials, in addition to electrical properties, resins with excellent heat resistance and low coefficient of thermal expansion are used, especially thermosetting resins. Polyimide (PI), for example, is an example of such a resin, and we expect that the application range of PI will be further expanded by adding heat dissipation properties to PI. We have synthesized PI consisting of 1, 2, 3, 4-cyclobutane tetracarboxylic dianhydride (CBDA) and 2, 2-bis[4-(4-aminophenoxy)phenyl]propane (BAPP) [52]. The PI matrix is not heat-dissipating by itself, so new PI hybrids can be fabricated by adding heat-dissipating inorganic materials such as magnesia (MgO). In 2018, Hara et al. [53] prepared a PI/MgO hybrid material with MgO finely dispersed in a PI matrix cross-linked with APrTMOS. Magnesium ethoxide (MgOEt) was used as a precursor for MgO and the MgO particles prepared in situ using the sol-gel reaction of APrTMOS-terminated PAA and MgOEt. MgO particles have high thermal diffusivity. The dispersion of MgO particles in the matrix creates thermal conduction pathways in the direction of the thickness of the hybrid film. In the hybrids prepared with 20 wt.% of the MgO hybrids, the thermal diffusivity and thermal conductivity values were approximately twofold higher than as those of PI alone.

Currently, there is a wide variety of silane coupling agents; however, the polymer matrix and type of metal alkoxide that can be dispersed depend on the coupling agent used. In some cases, hybrid properties can be improved by promoting the formation of crosslinks in the hybrid material, whereas, in other cases, the hybrid properties can be utilized using nanodispersed inorganic materials. Therefore, it is important to select an appropriate structure, such as a 3D network, according to the desired functionality required for the desired application. In the sol-gel method, a combination of several metal alkoxides can impart various properties to the material. Conversely, an increase in the number of inorganic components in the polymer matrix can affect the conformation of the polymer chains, and although the material gains new functionalities, it also loses the original properties of the organic polymer, such as formability, flexibility, and toughness.

Subsequently, the interactions formed between the polymer matrix and inorganic materials in the hybrid material, the particulate dispersion, and composite formation in the hybrid material derived from these interactions need to be considered.

## 4. Birth of Ionic Liquids, from Designer Solvents to Functional Ionic Liquids

Recently, ionic liquids have attracted considerable attention as a "third solvent," similar to organic solvents and water. Ionic liquids have a long history and have been the subject of numerous review articles [54,55,56,57,58,59,60]. They are generally defined as salts with melting points below 100 °C. The first ionic liquids were molten salts reported by Gabriel and Weiner [61] in 1888. In 1914, Walden [62] reported that ethylammonium nitrate is a molten salt with a melting point of 12 °C. This is the original form of today’s ionic liquids. They have subsequently attracted attention as functional liquids, mainly in the field of electrochemistry, and considerable research attention has been turned on ionic liquids with melting points near room temperature from the viewpoint of their convenience. Ionic liquids, which are also known as "designer solvents,” are available in many varieties due to the number of cation and anion species available. As organic ions are also used, there is a high degree of freedom in terms of their molecular design. However, the complex interactions formed between ions and high ionic concentrations make it difficult to predict the properties of ionic liquids.

In 1992, Wilkes et al. [63] developed an ionic liquid using BF_4_^–^ as the anion, which was found to be a room-temperature molten salt that is more stable to water and air than conventional ionic liquids. Since then, the research interest in ionic liquids and number of reports have increased. The excellent properties of ionic liquids as substances (salts) have been utilized in a wide range of practical applications, such as solar cell electrolytes [64], antistatic agents [65], lubricants [66], and antibacterial agents [67]. In addition, ionic liquids are not only used as solvents, but also as additives and functional liquids. Recently, ionic liquids have been used as functional materials, such as magnetic ionic liquids [68,69,70,71].

Ionic liquids can dissolve solutes that are not soluble in common organic solvents. A milestone was reported in 2002 by Swatloski et al. [72], who used an ionic liquid containing 1-butyl-3-methylimidazolium cations ([C_4mim_]^+^) to dissolve cellulose, a natural polymer. Fukaya et al. [73] clarified the correlation between the structure and polarity of ionic liquids and reported that a group of ionic liquids using various phosphoric acid derivatives as the anion have high polarity (evaluated using the Kamlet–Taft parameter [74]), which were stable and dissolved cellulose without heating. In particular, cellulose forms strong intra- and intermolecular hydrogen bonds. Therefore, it is advisable to design ionic liquids with high hydrogen bond-donating properties (donor type; the b-value used in the Kamlet-Taft parameter), which are effective for hydrogen bond breaking (relaxation) [75]. Abe et al. [76] reported that ionic liquids selectively dissolve cellulose when dissolved in an untreated biomass resource, such as bran. Bran is a macromolecular complex comprising cellulose, hemicellulose, and lignin, in which lignin forms a 3D network structure [75]. Therefore, lignin is not easily dissolved by the high polarity alone, but by appropriately controlling the polarity of the ionic liquid so that only cellulose can be isolated and extracted from bran. In other words, the design of the ionic liquid showed that it was highly selective for penetration into the bran matrix.

The affinity of ionic liquids toward polymeric materials is not limited to natural polymers such as cellulose; they also exhibit affinity toward synthetic polymers. The high compatibility and selectivity of ionic liquids with polymers can lead to the creation of new materials using ionic liquids as solvents and/or additives in the polymer matrix. Ueki et al. [77] reported the diverse compatibilities of various polymers using a hydrophobic ionic liquid ([C_2min_][NTf_2_]). PMMA is miscible with [C_2min_][NTf_2_] in any ratio and the in situ polymerization of the ionic liquid with MMA and a crosslinker enables the preparation of gels loaded with the ionic liquid. Aqueous solutions of poly(N-isopropylacrylamide) (PNIPAm) have a lower critical solution temperature (LCST) [78] of ~304 K, which is also observed in the volume phase transition of PNIPAm gels. By changing the solvent used in this system to [C_2min_][NTf_2_], the phase separation behavior follows a completely opposite trend, with PNIPAm + [C_2min_][NTf_2_] exhibiting an upper critical solution temperature (UCST). Conversely, polybenzyl methacrylate (PBnMA) shows an LCST that is soluble at low temperature, but insoluble at higher temperature. Scott et al. [79] reported that the addition of an ionic liquid ([C_4min_][PF_6_]) to PMMA, a polymeric material with a high glass transition temperature (*T*_g_ ), acts as a plasticizer to lower the *T*_g_. It was reported that the addition of [C_4min_][PF_6_]) to the PMMA matrix improves the thermal stability of PMMA by imparting the properties of the ionic liquid to PMMA.

## 5. Ionic Liquids/Collaborations with Hybrid Materials

The addition of inorganic substances to polymers significantly reduces their polymer-specific properties, such as formability and toughness. This is because the dispersion of inorganic substances on a nanometer scale results in strong interactions (e.g., hydrogen bonds and coordination bonds) between the inorganic components and the polymer, which unintentionally increases the cross-linking density of the matrix. However, if the interaction between inorganic components and polymers is completely inhibited, the freely behaving inorganic components will aggregate in the matrix. Therefore, the cross-linking density of the inorganic components and polymers must be controlled.

Hara [80] synthesized several ionic liquids and investigated the dispersion and stability of inorganic particles by ionic liquids. In general, the Hildebrand solubility parameter (SP value) is used as a guide when selecting a solvent for a polymer. However, the SP value concept is only valid for systems with weak interactions, such as London dispersion forces, and is often not applicable in highly polar solvents. In the case of polymers, the solubility differs depending on the polymer concentration and molecular weight distribution. In addition, the estimation of the SP values of ionic liquids is difficult because they are mainly based on Coulombic interactions, which often do not match the actual system. In such a situation, we found an ionic liquid, tetrabutylphosphonium chloride (TBPC; P[4,4,4,4]Cl), which specifically inhibited the hydrogen bonds formed between the polymer and inorganic components. When nanoparticles, such as titania, are added to PMMA, the hydrogen bonding between the C=O groups in PMMA and the OH groups present on the titania surface is enhanced. Hara et al. [81] reported that upon loading TBPC onto a hybrid material, only the hydrogen bonds were inhibited, and the hybrid film became flexible (Figure 4).

Due to its high affinity for PMMA, TBPC has no effect on the transparency in the visible light region. The cross-linking density of the PMMA hybrid material can also be adjusted by simply adjusting the MSi content; the addition of TBPC reduced the cross-linking density due to hydrogen bonding, resulting in a significant increase in the shape-memory performance (fourfold higher deformation rate) and toughness (28-fold higher) when compared to the hybrid prepared without TBPC. TBPC also reduced the cross-linking density of the material. Furthermore, the inhibitory effect of TBPC on the hydrogen bonding and thermal dissociation of the matrix can be controlled in the presence of TBPC [82]. The incorporation of titania and thermally dissociable carboxyl groups into the network structure of the hybrid can be used to prepare hybrid materials with reversible cross-linking, in which the cross-linked structure is retained at low temperature and dissociated upon increasing temperature; the cross-linked structure is reformed upon decreasing the temperature (Figure 5). In other words, a melt-moldable network polymer was created for the first time using organic–inorganic hybrid materials prepared via the sol-gel method.

Notably, the selectivity of the ionic liquid to act on the polar sites in the PMMA matrix enables the hybrid material to be melt-molded. Furthermore, the original properties of the matrix, such as the toughness and high transparency of PMMA, were retained. Titania did not aggregate before and after melt-molding and retained its UV-shielding ability, and the shape-memory properties of PMMA were maintained due to the hydrogen bond-inhibiting and plasticizing effects of TBPC. In other words, ionic liquids have the effect of balancing the basic properties of the constituents. This has enabled hybrid materials to evolve into highly functional materials that combine the superior properties of their incorporated components.

## 6. Creation of Functional Hybrids in the Presence of Ionic Liquids

The functional manifestations of hybrid materials prepared using the sol-gel method have been described. It is clear that the hybrid properties are highly related to the network structure of the matrix and dispersibility of the inorganic material. Here, we discuss the future prospects for the creation of functional organic–inorganic hybrid materials from the viewpoint of their dispersion in the matrix. The durability of functional nanoparticles in the matrix is particularly important for the preparation of functional hybrid materials and, from the viewpoint of their functional properties, the nature of the composite materials will evolve from the homogeneous dispersion systems currently used to dispersion-controlled systems in the future. Aida et al. [83] reported that UV irradiation of titania photoinduced a switch from Ti(IV) to Ti(III), and that the magnetic properties of titania switched from antimagnetic to paramagnetic. Despite its excellent magnetic properties, Ti(III) is easily oxidized in the presence of oxygen and cannot withstand long-term use, which is a new problem. Hara et al. [84] exploited the packing effect of the cyclohexane skeleton to control the oxygen permeability. The lifetime of Ti(III) in polycarbonate (PC), which does not have a cyclohexane skeleton, was ~2 h, but by nanodispersing titania in PC with a cyclohexane skeleton, they succeeded in extending the lifetime to ~10 d, which is 120 times longer than previously obtained. Furthermore, by covering the PC/titania hybrid film with a photomask and irradiating it with ultraviolet light at 250 W for 2 h, only the areas exposed to light were photoinduced to change to Ti(III) and become paramagnetic. The film is expected to be used as a flexible magnetic lithography substrate because it enables flexible patterning of the magnetic materials and magnetization (Figure 6).

However, the sol-gel method also faced the problem that the high dispersion and random orientation of titania did not allow a sufficient magnetic susceptibility to be achieved for use as a magnetic lithography substrate. Therefore, it is necessary to develop new substrates with aligned Ti(III) magnetic domains. In other words, the magnetic nanoparticles require the development of technology to control the phase separation process during the dispersion of the nanoparticles, i.e., the magnetic properties need to be improved by clustering the nanoparticles together. Ikake et al. [85] have developed a method to control the phase separation by combining the photoinduced phase transition and shape-memory processes (Figure 7).

We found that UV irradiation of the highly dispersed magnetic ionic liquid TBP [FeCl_4_] in a polymer matrix causes phase separation of the FeCl_3_ particles, which coincidentally align along the polymer chain when the polymer chain is uniaxially stretched. Hybrids with aligned particles exhibited a vivid color. When the hybrid material was irradiated with UV light without the shape-memory process, the hybrid material was colored brown by the FeCl_3_, but the magnetic susceptibility did not increase. By incorporating the aforementioned shape-memory property, the magnetic nanoparticles were not only aligned inside the hybrid material, but the magnetic domains were also aligned, and the magnetic susceptibility increased threefold when compared with the hybrid material in which FeCl_3_ was uniformly dispersed. Furthermore, partial patterning was possible when the hybrid film was covered with a photomask and irradiated with ultraviolet light and the shape-memory process carried out with only the UV-irradiated areas showing paramagnetism. This is the first reported attempt toward the preparation and morphological control of organic–inorganic hybrid materials via the combination of a photoinduced phase transition and shape-memory process. This knowledge is of great importance in the creation of organic–inorganic hybrid materials. In contrast to conventional hybrid materials, which focus on the highly homogeneous dispersion of inorganic particles in a polymer matrix, in this system, the hybrid material is prepared by promoting the microphase-separated structure of the particles. In particular, the use of the shape-memory property, which is also a characteristic of network polymers, and the use of the hybrid structure itself as a reaction field, are very unique, even though they are the product of chance.

## 7. Conclusions

This paper summarizes the creation of organic–inorganic hybrid materials as a historical fact. Various techniques and dispersion types were discussed, including in situ formation of inorganic particles using metal alkoxides, physical dispersion and dispersion by chemical bonding, but all have in common the creation of hybrid materials with a highly homogeneous dispersion in a polymer matrix. It is precisely for this reason that organic–inorganic hybrid materials have developed into essential materials with high performance and functionality. From a simple blend of organic and inorganic materials, a completely new material has emerged that is neither organic nor inorganic. Recently, the incorporation of ionic liquids into the field of organic–inorganic hybrid materials has revealed that ionic liquids are deeply involved in the control of the microstructure of hybrid materials, from their role as solvents to their selective action on polar sites. The use of functional ionic liquids as hybrid raw materials has also dramatically improved the dispersibility of inorganic nanoparticles down to the single-particle level, enabling the creation of highly nanodispersed hybrid materials. Functional ionic liquids have made it possible to functionally control 3D network structures and nanodispersion of fine particles. However, there are still unexplored areas. Biomimetic hybrid materials such as those described above are still some way off. Further research is awaited into the formation of dense structures corresponding to their functions and the control of their structure. Biomimetic hybrid materials are fabricated using only natural energy. Methods that actively utilize natural energy for the creation of new materials will be increasingly needed in the future. Only by solving these problems will organic–inorganic hybrid materials be transformed into novel functional materials that go beyond the composite rule.

## Figures and Tables

**Figure 1 polymers-14-03247-f001:**
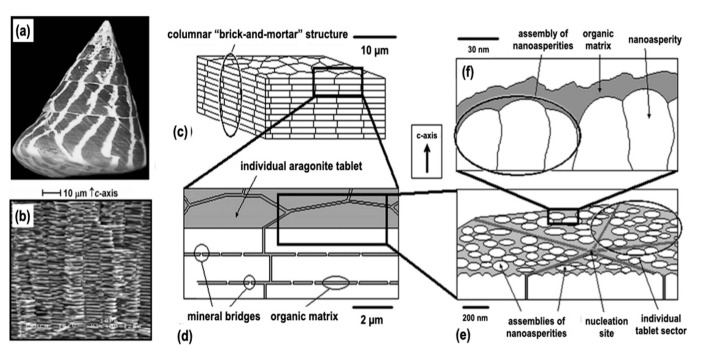
(**a**) Photograph of *Trochus niloticus* and (**b**) lateral-view SEM image of the internal stacked columnar nacreous layer microstructure of a fresh sample of *Trochus niloticus*. A schematic representation of the multi-scale hierarchical structure of the nacreous layer: (**c**) 10 μm, (**d**) 2 µm, and (**e**) 200 nm length scale of the individual tablet features, and (**f**) 30 nm length scale showing the nanoasperity structure. From ref. [11]. Adapted with permission from Ref. [11]. Copyright 2005, Materials Research Society.

**Figure 2 polymers-14-03247-f002:**
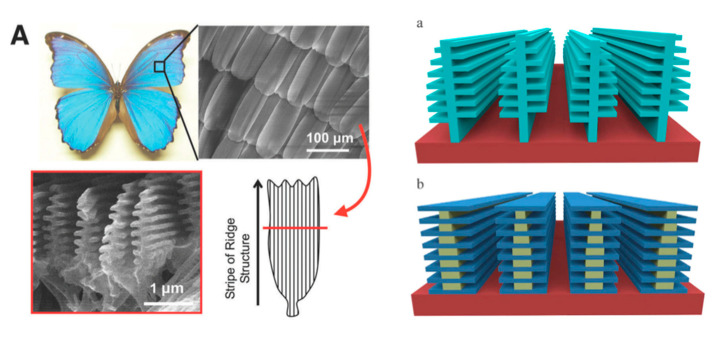
(**A**) Schematic representation of the scales of morpho butterfly wings. (**a**) Christmas tree structure of lamellar layers similar to actual wing scales. (**b**) Designed scales to be fabricated with aligned lamellae structures of PMMA/LOR (LOR; lift-of-resist supplied by MicroChem Corp.) alternate layers. From ref. [12].

**Figure 3 polymers-14-03247-f003:**
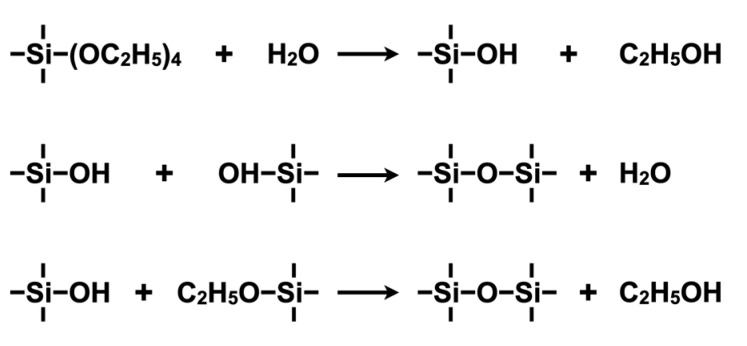
Key chemical reactions for the hydrolysis and polycondensation of tetraethyl orthosilicate (TEOS) under acidic or basic conditions. This is the central basic reaction in the sol-gel method, from which many functional hybrids are prepared.

**Figure 4 polymers-14-03247-f004:**
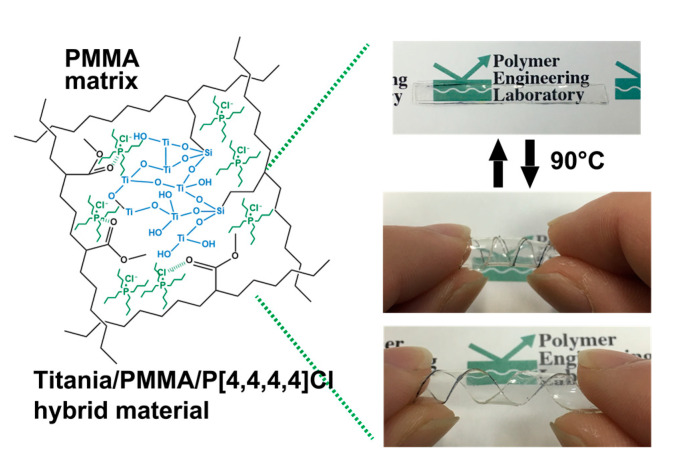
Schematic representation of the hybrid material composed of poly(methyl methacrylate) (PMMA), titania, and tetrabutylphosphonium chloride (TBPC). The transparency, toughness, and shape memory of the PMMA/titania hybrid are improved.

**Figure 5 polymers-14-03247-f005:**
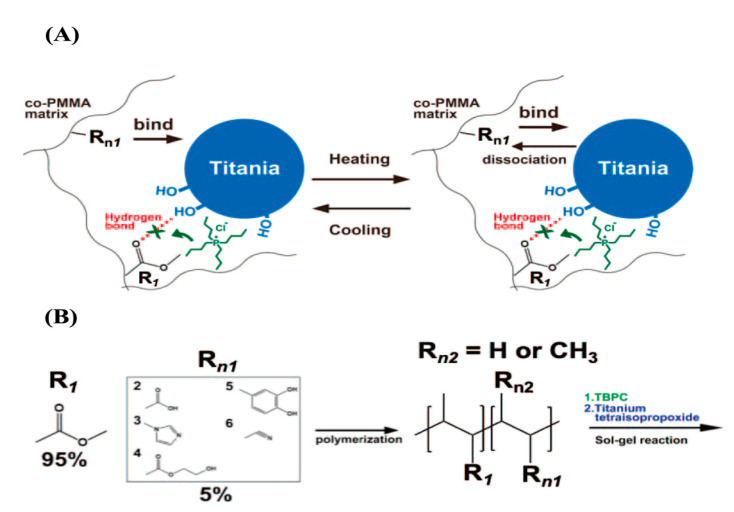
(**A**) TBPC inhibits hydrogen bonding between the carbonyl group of MMA (R1) and the hydroxy groups on the surface of titania particles; the functional group (Rn) of the other candidate monomer is thermally reversibly bound and unbound in the PMMA copolymer/titania hybrid material. (**B**) Monomers with functional groups capable of bonding to the surface of titania particles; the functional group (Rn) of the other candidate monomer is thermally reversibly bound and unbound in the PMMA copolymer/titania hybrid material. Monomers with functional groups capable of radical polymerization: R2 is MA, R3 is VI, R4 is HE, R5 is DMA, and R6 is AN. From ref. [82]. Reprinted with permission from Ref. [82]. Copyright 2020 American Chemical Society.

**Figure 6 polymers-14-03247-f006:**
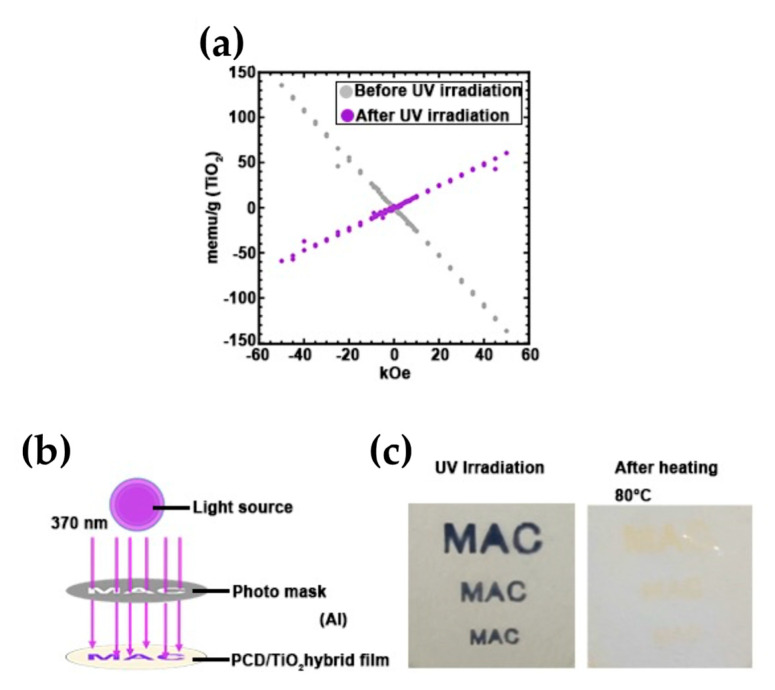
(**a**) Superconducting quantum interference device (SQUID) magnetometry measurements of PC100: before (grey plot) and after (purple plot) UV irradiation. (**b**) Image of the photolithography of PC100 using a UV light source. (**c**) Photograph of PC100 upon photolithography and heat treatment.

**Figure 7 polymers-14-03247-f007:**
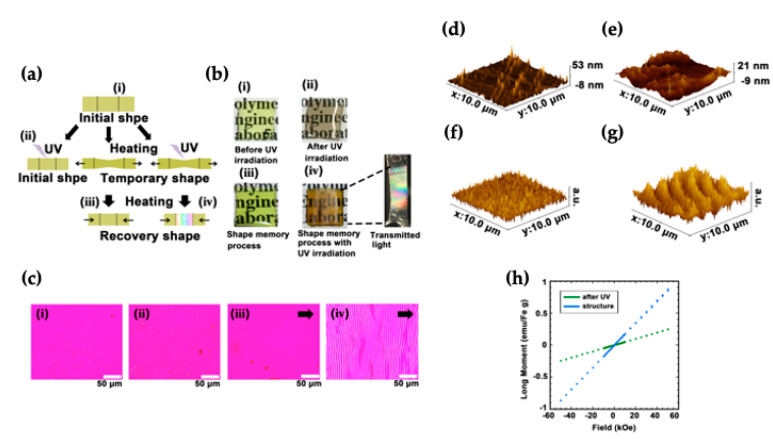
(**a**) Preparation of FeCl_3_-oriented hybrid films. (**b**) Photographs of Ti15TB40Fe5 after the (iii) shape memory and (iv) UV irradiation and shape-memory processes. (**c**) Photographs of Ti15TB40Fe5 after each process and state, with the image (iv) expanded to show a photograph of the transmitted light. The white bar represents 50 mM. AFM images of each state of Ti15TB40Fe5 after (**d**) UV irradiation and (**e**) UV irradiation and the shape-memory process. MFM images of each state of Ti15TB40Fe5 after (**f**) UV irradiation and (**g**) UV irradiation and the shape-memory process. (**h**) Magnetometry measurements of Ti15TB40Fe5 after UV irradiation (green plot) and the shape-memory process (blue plot). From ref. [85]. Adapted with permission from Ref. [85]. Copyright 2022, The Royal Society of Chemistry.

## Data Availability

Not applicable.

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
