# Peer review of "Skillful Control of Dispersion and 3D Network Structures: Advances in Functional Organic–Inorganic Nano-Hybrid Materials Prepared Using the Sol-Gel Method"

_polymers, 2022, doi:10.3390/polym14163247_

Round 1

Reviewer 1 Report

In this manuscript, the authors reviewed the development of organic-inorganic nanohybrid material prepared using sol-gel method. The manuscript is well-written, I have some minor suggestions.

1. I suggest the authors to compare the sol-gel methods with other methods that are used to prepare organic-inorganic nanohybrids, and highlight the advantages of the sol-gel method over the other methods. For example, solution based self-assembly ("Core/shell conjugated polymer/quantum dot composite nanofibers through orthogonal non-covalent interactions." Polymers 8.12 (2016): 408.; Hybrid conjugated polymer/magnetic nanoparticle composite nanofibers through cooperative non-covalent interactions." Nanoscale Advances 2.6 (2020): 2462-2470.), melt-blending ("HLDPE/organic functionalized SiO2 nanocomposites with improved thermal stability and mechanical properties." Composite Interfaces 11.8-9 (2005): 687-699.; "A critical review on the manufacturing processes in relation to the properties of nanoclay/polymer composites." Journal of Composite Materials 47.9 (2013): 1093-1115.), etc. 

2. I suggest the authors to also summarize the common applications of organic-inorganic hybrids prepared by sol-gel methods. 

3. I suggest the authors to add a perspective or future direction section, for example what are the aspects that still need development in this field 

Author Response

Review1

Answer to Reviewer 1       

  1. I suggest the authors to compare the sol-gel methods with other methods that are used to prepare organic-inorganic nanohybrids, and highlight the advantages of the sol-gel method over the other methods. For example, solution based self-assembly ("Core/shell conjugated polymer/quantum dot composite nanofibers through orthogonal non-covalent interactions." Polymers8.12 (2016): 408.; Hybrid conjugated polymer/magnetic nanoparticle composite nanofibers through cooperative non-covalent interactions." Nanoscale Advances2.6 (2020): 2462-2470.), melt-blending ("HLDPE/organic functionalized SiO2 nanocomposites with improved thermal stability and mechanical properties." Composite Interfaces 11.8-9 (2005): 687-699.; "A critical review on the manufacturing processes in relation to the properties of nanoclay/polymer composites." Journal of Composite Materials 47.9 (2013): 1093-1115.), etc. 

→I think there is a point of view that you have pointed out. However, in this paper, the focus is not on the superiority of the sol-gel method over other methods, but on the historical facts about the development of organic-inorganic hybrid materials. In particular, we have carefully scrutinised the papers on the realisation of highly homogeneous dispersion and the formation of 3D network structures in polymer matrices, on which much emphasis has been placed so far, so that it is easy to follow the starting point and the changes in thinking over time. In the course of this research, problems such as a reduction in the flexibility of the matrix and poor formability and workability arose as a result of compositing a large number of inorganic particles, and the emphasis has now shifted from the formation of the network structure of hybrid materials to the control of the network structure. The lack of a summary of the transition of the network structure in organic-inorganic hybrid materials has led to the publication of this paper.

  1. I suggest the authors to also summarize the common applications of organic-inorganic hybrids prepared by sol-gel methods. 

→General examples of organic-inorganic hybrid materials wre mentioned together with references. However, as stated in point 1, our claims are historical facts concerning the development of organic-inorganic hybrid materials, with particular reference to the highly homogeneous dispersion of hybrid materials and the formation of 3D network structures. In particular, the concept of network structures has changed over time. As can be seen from the starting point of the paper, emphasis was initially placed on dispersion in the polymer matrix. However, this led to the application of silane coupling agents because of their low dispersibility, but this led to a new problem: the flexibility, formability and workability of polymers were reduced when a large amount of inorganic particles were dispersed. For this reason, the emphasis has now shifted to controlling rather than forming 3D network structures. However, although there are many papers on the functions of hybrid materials, few of them focus on the 3D network structure and introduce it in line with the changing times, which is why we decided to introduce it in this paper. For this reason, it was decided to limit the application examples to an introduction only in order to organise the discussion points.

  1. I suggest the authors to add a perspective or future direction section, for example what are the aspects that still need development in this field 

→We believe that you are right. We have organised the findings of our research and modified them to be mentioned in the conclution section.

Reviewer 2 Report

In this review, the authors discussed the organic-inorganic hybrid materials which have become indispensable high-performance and highly functional materials. This is owing to the improved dispersion control in hybrid materials and the emergence of functional ionic liquids. Harmonization of both these factors has enabled the utilization of functional 3D network structures and nanodispersions in composite materials. The authors summarized the historical development of hybrid materials prepared using the sol-gel method and the birth of ionic liquids. Also, they presented the historical results leading to the development of functional 3D network structures and dispersion control, as well as a review of the research on functional ionic liquids. The interpretations and conclusions are well justified by the results. However, the quality of the figures needs to be improved and the references are not up to date (20% present last 10 years). We believe that this research subject is promising for the use of functional ionic liquids as hybrid raw materials. The dispersibility and the inorganic nanoparticles have been refined and improved on a single-particle level, resulting in highly nanodispersed hybrid materials. Organic-inorganic hybrid materials are subject to interesting points and will be transformed into novel functional materials beyond the composite rule.

Summary: I recommend publishing this manuscript after considering my comments on the attached file.

Author Response

Review2

In this review, the authors discussed the organic-inorganic hybrid materials which have become indispensable high-performance and highly functional materials. This is owing to the improved dispersion control in hybrid materials and the emergence of functional ionic liquids. Harmonization of both these factors has enabled the utilization of functional 3D network structures and nanodispersions in composite materials. The authors summarized the historical development of hybrid materials prepared using the sol-gel method and the birth of ionic liquids. Also, they presented the historical results leading to the development of functional 3D network structures and dispersion control, as well as a review of the research on functional ionic liquids. The interpretations and conclusions are well justified by the results. However, the quality of the figures needs to be improved and the references are not up to date (20% present last 10 years). We believe that this research subject is promising for the use of functional ionic liquids as hybrid raw materials. The dispersibility and the inorganic nanoparticles have been refined and improved on a single-particle level, resulting in highly nanodispersed hybrid materials. Organic-inorganic hybrid materials are subject to interesting points and will be transformed into novel functional materials beyond the composite rule.

Summary: I recommend publishing this manuscript after considering my comments on the attached file.

Answer to Reviewer 2

[1] Reference should be added.

→References added and description amended.

 Hara [79] synthesised several ionic liquids and investigated the dispersion and stability of inorganic particles by ionic liquids.

[2] The resolution of the PMMA matrix should be improved.

→Figure has been replaced.

[3] The density of the materials reduces by TBPC, why? what is your evidence?

→Errors in the description. The “density” changed to the “cross-linked density”.

 TBPC also reduced the crosslinking density of the material.

[4] what is the irradiated dose?

→Information on irradiated ultraviolet radiation has been added.

 Furthermore, by covering the PC/titania hybrid film with a photomask and irradiating it with ultraviolet light at 250 W for 2 h, only the areas exposed to light were photoinduced to change to Ti(III) and become paramagnetic.

[5] The resolution of the figure should be improved.

→Figure has been replaced.

[6] The resolution of the figure is poor. It should be high resolution within 600 dpi.

→Figure has been replaced.

[7] 20% of the references are within last ten years, it is better to up date to be included recently references.

→In this paper, we have focused on the creation of hybrid materials using the sol-gel method, and the homogeneous dispersion and 3D network construction, which are the characteristics of the resulting hybrid materials, in order to provide the reader with historical facts. Silane coupling agents play an important role in this method, and it is necessary to consider the history of the birth of silane coupling agents and the transition of hybrid materials. Silicon is the only material that exists abundantly in nature and binds stably to the organic phase (carbon), and organic-inorganic hybrid materials have been developed by making use of this. On the other hand, when the amount of inorganic composites is increased, a strong network is formed, making the hybrid material hard and brittle, and the original properties of polymers are lost. This paper aims to inform about the causes of this and how to deal with it. There is a lot of research on performance-specific hybrid materials, and I am convinced that the re-conceptualisation of the design basis of composite materials, not just the glamorous aspects, will lead to the development of hybrid materials in the future. We have presented the papers that are the starting point for hybrid materials, and from there we have considered the future through our research. For this reason, we have chosen to cover a number of papers that are sufficiently widely cited rather than the most recent data.

[8] Should be completed?

→Page numbers have been corrected.

 568/1-568/9.

Author Response

Review

Answer to Reviewer 3

  1. The authors should describe some applications of the distinct types of hybrid materials described, concerning the applicability of the achieved properties/functionalities as devices, such as sensors, energy harvesting, filters, etc.

→This paper describes the historical facts concerning the construction of 3D network structures and the highly homogeneous dispersion of organic-inorganic hybrid materials. We have therefore presented our results as an example of the developmental transition of hybrid materials. For this reason, we consider the application examples of the hybrid materials pointed out to be of little importance and have left them as they were in the original text. However, energy harvesting is a very important point and an unexplored and urgent issue in the development of hybrid materials, so we have described it in the summary section as the most important theme for the future, based on the points raised.

  1. In my point of view, I believe that the manuscript lacks recent references. There are only a few references from the last 4 years.

→In this paper, we have focused on the creation of hybrid materials using the sol-gel method, and the homogeneous dispersion and 3D network construction, which are the characteristics of the resulting hybrid materials, in order to provide the reader with historical facts. Silane coupling agents play an important role in this method, and it is necessary to consider the history of the birth of silane coupling agents and the transition of hybrid materials. Silicon is the only material that exists abundantly in nature and binds stably to the organic phase (carbon), and organic-inorganic hybrid materials have been developed by making use of this. On the other hand, when the amount of inorganic composites is increased, a strong network is formed, making the hybrid material hard and brittle, and the original properties of polymers are lost. This paper aims to inform about the causes of this and how to deal with it. There is a lot of research on performance-specific hybrid materials, and I am convinced that the re-conceptualisation of the design basis of composite materials, not just the glamorous aspects, will lead to the development of hybrid materials in the future. We have presented the papers that are the starting point for hybrid materials, and from there we have considered the future through our research. For this reason, we have chosen to cover a number of papers that are sufficiently widely cited rather than the most recent data.

  1. On figure 1, the authors should verify the species name. it seems that it was mistakenly written.

→The species names in Figure 1 have been changed.

  1. All the mechanical properties described in lines 55 to 58 seem very interesting, although the reader is not able to verify this. The authors should add comparison values.

→Although we consider it an interesting property and a numerical value, it is important that a dense structure is formed as a biomimetic hybrid material and the numerical value is not significant. Therefore, this information is unnecessary and has been deleted.

  1. Figure 2 seems somehow out of place. The butterfly is not clearly mentioned in the text. The authors should verify this.

→The nanophotonic structure of the wings of the morpho butterfly is a very important indicator for the creation of organic-inorganic hybrid materials, especially for structural control. The fabrication of dense structures is also an outstanding advantage of biomimetic hybrid materials. This is still difficult to achieve in organic-inorganic hybrid materials. However, it is pointed out that this information is not given in the text and has been corrected in the addition. In addition, the precise control of structure is a very important topic in the creation of future hybrid materials and has been added in the summary section for the reader to understand.

  1. Lines 144-145 seem to be repeating a previously mentioned idea.

→This is a duplicate of the previous paragraph. The following sentence has been deleted.

"In particular, nanometer-sized primary particles comprising the inorganic materials are expected to exhibit unique optical, electrical, and magnetic properties that differ from those of their bulk materials. ."

The wording was also changed for lines 139 to 142.

  1. The text concerning PU appears to be unconnected to previous text. The authors should try to make this connection clearer.

→The significance and efficacy of the network structure of PU hybrid materials are organised and the description is revised. It is clearly stated that the network structure contributes not only to the mechanical properties, but also to the stability of the material.

 The 3D network structure not only gives the hybrid material flexibility, but also contributes to the weather resistance of the material. One of the general-purpose polymers is polyurethane (PU), a generic term for polymers with urethane (-NHCOO-) groups in the main chain structure, obtained by the polyaddition reaction of polyhydric alcohols and isocyanates. Although PU exhibits excellent rubber elasticity due to the pseudo-network structure induced by isocyanates, aromatic isocyanates are vulnerable to light exposure, causing yellowing and oxidative degradation. Ikake et al [42, 43] focused on the network structure of PU and used a light-stable cross-linking agent instead of aromatic isocyanates to construct a 3D. The properties of PU elastomers derive from this network structure, which also offers the advantage that the crosslink density can be easily adjusted by adding appropriate amounts of metal alkoxides. Both ends of the polyol were modified with silane coupling agents and titanium tetraisopropoxide was added to generate titania nanoparticles in situ, forming a 3D network within the PU matrix. The silane coupling agent enabled the titania nanoparticles to be highly dispersed in the PU matrix, resulting in hybrids with excellent transparency in the visible light range and UV-protection properties without yellowing. Furthermore, by changing the type of metal alkoxide used, various properties can be imparted to the PU.

  1. A reference is missing in lines 249-250.

→Line 249 describes the research report in reference [51], which was left in the original. Line 250, on the other hand, is our earlier work, with the description corrected and the reference added.

  1. I believe that the manuscript, in general, need more and improved figures to illustrate the cited examples.

→In this paper, reference was made to the preparation of organic-inorganic hybrid materials, in particular 3D network structures and highly homogeneous dispersion. Our research results were discussed as an example of this transition. The idea of microstructures in hybrid materials has changed over time, and the creation of network structures was a challenge in the hybrid materials with silane coupling agents discussed in Class II. However, it was found that the flexibility and formability of polymers decreased as more inorganic particles were composited, and today the emphasis has shifted from the formation of network structures to their control. Figures necessary to illustrate this transition have been included, but information that is different from the subject matter has been cut. Instead, to make it easier to follow the transition in the development of hybrid materials, many papers that are necessary as historical facts, such as papers that serve as starting points and papers that trace the transition in time, have been introduced.